# Insights into Potential Pathogenesis and Treatment Options for Immune-Checkpoint Inhibitor-Related Pneumonitis

**DOI:** 10.3390/biomedicines9101484

**Published:** 2021-10-16

**Authors:** Hiroyuki Ando, Kunihiro Suzuki, Toyoshi Yanagihara

**Affiliations:** 1Research Institute for Diseases of the Chest, Graduate School of Medical Sciences, Kyushu University, Fukuoka 812-8582, Japan; ando.hiroyuki.781@m.kyushu-u.ac.jp (H.A.); suzuki.kunihiro.784@m.kyushu-u.ac.jp (K.S.); 2A Department of Respiratory Medicine, Hamanomachi Hospital, Fukuoka 810-8539, Japan

**Keywords:** immune checkpoint, immune-checkpoint inhibitor-related pneumonitis, immune-related adverse events (irAEs)

## Abstract

Immune-checkpoint inhibitors (ICIs) targeting cytotoxic T-lymphocyte antigen 4 (CTLA-4), programmed cell death-1 (PD-1), and programmed cell death-1-ligand 1 (PD-L1) have become new treatment options for various malignancies. ICIs bind to immune-checkpoint inhibitory receptors or to the foregoing ligands and block inhibitory signals to release the brakes on the immune system, thereby enhancing immune anti-tumor responses. On the other hand, unlike conventional chemotherapies, ICIs can cause specific side effects, called immune-related adverse events (irAEs). These toxicities may affect various organs, including the lungs. ICI-related pneumonitis (ICI-pneumonitis) is not the most frequent adverse event, but it is serious and can be fatal. In this review, we summarize recent findings regarding ICI-pneumonitis, with a focus on potential pathogenesis and treatment.

## 1. Introduction

Immune-checkpoint inhibitors targeting cytotoxic T-lymphocyte antigen 4 (CTLA-4), programmed cell death-1 (PD-1), and programmed cell death-1-ligand 1 (PD-L1) have provided substantial benefits to many cancer patients [1,2,3,4]. ICIs utilize the immune response to tumor antigens by activating T cells [5,6]. However, these agents also target the same pathways that maintain immune tolerance and prevent autoimmunity, inducing immune-related adverse events (irAEs), including pneumonitis, hypophysitis, diabetes, colitis, thyroiditis, and others [7,8,9,10,11]. Importantly, ICI-pneumonitis represents the most common fatal irAE from anti-PD-1/PD-L1 monotherapy, accounting for 35% of anti-PD-1/PD-L1-related deaths [12]. In this paper, we summarize recent evidence for ICI-pneumonitis, focusing on possible pathobiology and treatment options.

## 2. Immune Checkpoints and Their Inhibitors

Optimal T cell activity requires binding of co-stimulatory molecules such as CD28 on the T cell surface to its ligands, CD80 (B7-1) and CD86 (B7-2), expressed on antigen-presenting cells (APCs) [13]. CTLA-4 is a transmembrane protein with structural similarities to CD28. It occurs on conventional T cells at a low basal level and on CD4^+^CD25^+^ regulatory T cells (T_reg_ cells), constitutively. CTLA-4 has a higher affinity for CD80/CD86 than CD28 and induces inhibitory signals in conventional T cells, resulting in the suppression of conventional T cell proliferation and activation [14]. CTLA-4 is essential for anti-inflammatory cytokine release from T_reg_ cells, which reduces the activation and proliferation of conventional T cells nearby. Further, Treg cells also prime APCs to induce the anergy of conventional T cells by internalizing and degrading CD80/CD86 ligands on APCs [15].

PD-1 is a transmembrane receptor in the immunoglobulin superfamily, found primarily on the surfaces of mature T cells [16]. PD-1 binds a specific ligand (PD-L1 or PD-L2) to suppress the activation of T cells. The blockade of the PD-1 axis by administration of an anti-PD-1 (or anti-PD-L1 or anti-PD-L2) antibody blocks ligand binding and unleashes anti-tumoral T cell activity by enhancing their effector functions and supporting the formation of memory cells. More T cells can subsequently bind to tumor antigens presented with MHC molecules on tumor cells via T cell receptors (TCRs). Consequently, this binding leads to the release of cytolytic mediators, such as perforin and granzyme, from activated T cells toward target cells, resulting in tumor elimination.

Cancer cells exploit checkpoint activation through these PD-1 and CTLA-4 pathways to induce anergy in antitumor T cells. The inhibition of CTLA-4 enhances T cell responses to tumor-associated neoantigens, depletes local intra-tumoral T_reg_ cells, and shifts the balance of the tumor microenvironment away from immunosuppression [17,18,19]. The PD-1/PD-L1 blockade releases the brakes on T cells and enhances T cell cytotoxicity toward tumor cells [20]. Ipilimumab, a human IgG1, anti-CTLA-4 monoclonal antibody (mAb), received FDA approval for the treatment of advanced melanoma in 2011. Pembrolizumab and nivolumab are human IgG4 anti-PD-1 antibodies, approved in 2014 by the U.S. FDA for the treatment of patients with refractory melanoma, and in 2015, for patients with advanced non-small-cell lung cancer. The first anti-PD-L1 antibody approved by the FDA was atezolizumab for urothelial cancers in 2016, followed by avelumab in 2017 for Merkel cell carcinoma and durvalumab for bladder carcinoma.

Unfortunately, a certain percentage of patients with cancers still fail to respond to therapies that target CTLA-4 and PD-1/PD-L1. The next wave of co-inhibitory receptors is LAG-3, TIM-3, and TIGIT. Although they belong to the same immunoglobulin family as PD-1 and CTLA-4, these receptors exhibit unique functions with specific tissue sites [21]. Now, target therapies for these receptors are being explored in clinical trials for cancer treatment.

## 3. Pathobiology of ICI-Pneumonitis

Detailed pathogenesis underlying irAEs remains largely unknown, but is thought to be related to the function that immune checkpoints serve in regulating immunological homeostasis [22]. There are four possible mechanisms: (1) Increasing T cell activity against antigens presented in healthy tissues, (2) increasing levels of preexisting autoantibodies, (3) increasing levels of inflammatory cytokines, and (4) enhancing complement-mediated inflammation related to the direct binding of anti-CTLA-4 antibodies to CTLA-4-expressing normal tissues [22].

Beyond these possible mechanisms, cellular immunity mediated by T lymphocytes is probably involved (Figure 1). We previously reported that T-cell-dominated lymphocytosis is evident in bronchoalveolar lavage fluid (BALF) from ICI-pneumonitis [23,24], which is consistent with other case series [25,26,27]. Proportions of CD8^+^ T cells expressing immune-checkpoint proteins (CD8^+^PD-1^+^TIM-3^+^), characteristic of tumor-infiltrating T lymphocytes (TILs) [28], are higher in ICI-pneumonitis than in other ICI-associated causes [23,24,29]. These observations suggest that TILs may contribute to the pathogenesis of ICI-ILD. Indeed, next-generation sequencing of the complementarity-determining region of the T cell receptor (TCR) identified a substantial number of T cell clones in BALF among T cells in peritumoral lesions [30]. In addition, many tumor-associated clones with a read frequency of ≥0.10% were also present in BALF. Another recent study showed overlapping clonality in lesions of ICI-pneumonitis (due to pembrolizumab) and in the primary tumor site [31]. These facts support the hypothesis that ICI-pneumonitis may be induced by ICI-activated T cells recognizing self-peptides or epitopes shared between the tumor and the self. Suresh et al. reported increased numbers of central memory T cells (high CD62L and low CD45RA expression), increased tumor necrosis factor (TNF)-α^hi^ interferon(IFN)δ^hi^CD8^+^ T cells, and decreased Treg suppressive phenotypes in BALF from ICI-pneumonitis [27]. Presumably, these TNF-α^hi^IFNδ^hi^CD8^+^ T cells are the same population as CD8^+^PD-1^+^TIM-3^+^ detected in our study.

Intriguingly, the proportion of PD-1^+^PD-L1^+^ cells among CD8^+^ T cells in BALF was correlated with the severity of ICI-pneumonitis [24]. Recent studies revealed that antigen-presenting, cell-intrinsic PD-1 or CD80 interacts with PD-L1 in *cis* (on the same cell surface) to inhibit PD-1 signaling in T cells [32,33]. These facts suggest that *cis*-PD-1/PD-L1 interaction may suppress self-reacting T cells against lung tissue, and this disruption of *cis*-PD-1/PD-L1 regulation by ICIs can induce aberrant activation of self-reacting T cells, resulting in severe ICI-pneumonitis.

Recent evidence suggests that autoantibodies may be involved in the pathogenesis of ICI-pneumonitis, similar to skin and endocrine irAEs. Tahir et al. investigated autoantibodies in patients treated with ICIs using a technique called high-throughput serological analysis of recombinant cDNA expression [34]. The authors found increased levels of anti-CD74 autoantibodies in two patients with ICI-pneumonitis from a discovery cohort. This finding was verified in a confirmation cohort of 32 patients, 10 with and 22 without ICI-pneumonitis. A median 1.34-fold increase in anti-CD74 autoantibodies was detected when comparing ICI pre- and post-treatment in plasma. Further, levels of anti-CD74 autoantibodies, pre- and post-treatment, were significantly increased in patients with pneumonitis compared with those without pneumonitis [34]. CD74 is an intracellular chaperone molecule for major histocompatibility complex II and can be expressed on surfaces of immune cells, including macrophages [35]. Modest levels of CD74 were expressed in normal human lung tissue, while dramatically increased levels of CD74 were found in lung tissues from ICI-pneumonitis [34]. These facts indicate the pathogenic role of CD74 and its autoantibodies in the development of ICI-pneumonitis.

To better understand pathomechanisms, it may be necessary to investigate the imbalance between effector T cells and suppressor T cells (Tregs), to assess the degree of involvement of PD-1/PD-L1 *cis*-regulation in autoimmune tolerance, to discover self-antigens expressed in the lungs, and to assess the biological role of anti-CD74 autoantibodies. Recently, machine learning has been developed to predict TCR binding specificities of antigens presented by class I major histocompatibility complexes using only the TCR sequence, antigen sequence, and class I major histocompatibility complex allele [36]. This method can be applied to predict self-antigens in ICI-pneumonitis as well, which will unveil the genuine targets of ICI-pneumonitis.

## 4. Incidence and Risk Factors for ICI-Pneumonitis

The incidence of ICI-pneumonitis has been evaluated. The incidence of irAEs depends on tumor origin and the type of ICIs [37,38]. In clinical trials, ICI-pneumonitis is more common in patients with non-small cell lung cancer than in those with melanoma (4.1% vs. 2.7%) [39], and is more frequently reported in patients receiving ICI combination therapies, such as ipilimumab/nivolumab (10%) [40]. In clinical practice, ICI-pneumonitis developed more frequently, accounting for 7–19% of all patients [41,42]. Several risk factors for ICI-pneumonitis have been evaluated. These include underlying lung diseases (interstitial lung diseases, chronic obstructive pulmonary disease), performance status, prior thoracic radiotherapy, and the treatment combination of epidermal growth factor receptor (EGFR)-tyrosine kinase inhibitors with ICIs [42]. In particular, prior thoracic radiotherapy (odds ratio (OR) 3.33), prior lung disease (OR 2.82), and combination therapy (3.42) are identified as risk factors, analyzed by multiple logistic regression in patients with malignancies treated with ICIs [43]. These facts suggest a hypothesis that lung damage by radiation, chemotherapy, or existing lung diseases may result in denatured self-antigens in the lung, which attract circulating T cells that are activated by ICIs. Such activated T cells may misidentify these self-antigens as non-self, inducing ICI-pneumonitis.

## 5. Clinical Presentation of ICI-Pneumonitis

Clinical characteristics of ICI-pneumonitis patients vary, but dyspnea (53%), cough (35%), fever (12%), and chest pain (7%) have been reported [40]. The onset of ICI-pneumonitis varies, with a reported median onset of about 3 months [40,44]. It has been suggested that ICI-pneumonitis induced by combination therapies has an earlier onset than that od monotherapy [40]. Respiratory failure may progress rapidly, so caution is required. When respiratory symptoms appear, clinicians must assume diseases and conditions other than ICI-pneumonitis (Table 1). It is also necessary to understand the patient’s background, especially co-morbidities, oral medications, allergies, etc. First, infection should be kept in mind, and bacterial pneumonia, pneumocystis jirovecii pneumonia (PjP), and flare-ups of tuberculosis should be considered [45]. Prolonged high-dose steroid therapy and concurrent chemoradiotherapy were risk factors for PjP development among patients with lung cancer [46]. The reactivation of tuberculosis by immune checkpoint inhibitors has been reported [47,48,49]. In addition, sarcoid reaction and myasthenia gravis should be considered when respiratory deterioration suddenly occurs [49,50,51].

### 5.1. Radiographic Findings

A chest CT scan is important for the diagnosis of interstitial lung diseases (ILDs) and is also important for ICI-pneumonitis [52]. There have been various reports on image patterns of chest CT scans of drug-induced lung injury, but they differ depending on the drug administered [53]. Similarly, various image patterns have been reported in ICI-pneumonitis. ICI-pneumonitis is considered a lung injury from the acute phase to the organizing phase and fibrosis and presents four primary patterns: Organizing pneumonia (OP), nonspecific interstitial pneumonia (NSIP), hypersensitivity pneumonitis (HP), and diffuse alveolar damage (DAD) (Figure 2). The OP pattern is defined by peripheral consolidations and air bronchograms as the main features. The NSIP pattern is defined by several features, such as ground-glass opacities with traction bronchiectasis in peribronchovascular predominance. The HP pattern is defined by centrilobular nodules or a bronchiolitis-like appearance. The DAD pattern is characterized by scattered or diffuse areas of ground-glass opacities with or without traction bronchiectasis.

In our study, 77% OP, 15% HP, and 7.7% NSIP patterns were observed among 13 cases of ICI-pneumonitis [24]. Nishino et al. reported 20 cases of ICI-pneumonitis in 10 trials of nivolumab imaging with the following rates: 65% OP, 15% NSIP, 10% HP, and 10% DAD [54]. Myriam et al. reviewed imaging from 64 cases and reported the following rates: 23% OP, 15% HP, 6% NSIP, and 36% no classification [7]. Naidoo et al. reported 43 cases with imaging patterns of 37% ground-glass opacities, 22% HP, 19% OP, and 22% no classification [40].

### 5.2. Bronchoalveolar Lavage and Histology for ICI-Pneumonitis

Only a few papers have reported results of bronchoalveolar lavage (BAL) or lung biopsies. Generally, in other drug-related lung diseases, bronchoscopic evaluation (BAL with/without transbronchial biopsy) proves important not only in diagnosis, but also in the exclusion of other diseases, such as infection [55]. Similarly, in ICI-pneumonitis, bronchoscopy can be useful in the diagnosis and exclusion of other diseases, such as infection or tumor progression [52]. BALF in patients with ICI-pneumonitis reportedly shows lymphocytosis [7,24,56,57,58]. Further, at a research level, we have reported that immune-checkpoint profiles of T cells from BALF can be useful to distinguish ICI-pneumonitis from other lung diseases and may be useful for predicting disease severity [24].

Regarding histological examination, transbronchial lung biopsies should be recommended rather than surgical lung biopsies, according to the 2018 American Society of Clinical Oncology (ASCO) clinical practice guidelines [52]. Histological findings were numerous: Cellular interstitial pneumonitis, OP, DAD, acute fibrinous and organizing pneumonia (AFOP), and granuloma have been reported [40,50,57,59]. Altogether, these BAL and pathological results have been mostly consistent with CT scan findings, especially patterns of OP and NSIP, explaining the usual good response to corticosteroid therapy after discontinuing ICI treatment. Notably, it is necessary to determine the indication of bronchoscopy considering the risks and benefits, because it varies depending on the degree of respiratory failure and urgency of treatment.

## 6. Treatment for ICI-Pneumonitis

Several organizations, including ASCO, the European Society for Medical Oncology (ESMO), the Society for Immunotherapy of Cancer (SITC), and the Japanese Respiratory Society (JRS) have published formal guidelines for the management of ICI-pneumonitis [52,60,61,62]. Treatment of ICI-pneumonitis is based on the clinical severity of the pneumonitis. Toxicity severity is described using the Common Terminology Criteria for Adverse Events (CTCAE), on a scale of 1 (mildest) to 5 (death related to drug toxicity) (Table 2). Specifically for pneumonitis, grade 1 is defined by radiographic changes without symptoms. In these settings, withholding of ICI treatment is recommended, and patients should be monitored closely. Grade 2 pneumonitis is defined by radiographic infiltrates accompanied by mild-to-moderate symptoms, but without hypoxia. In these settings, pneumonitis can be often managed with corticosteroids. Intravenous prednisolone (PSL) 0.5–1.0 mg/kg/day or the oral equivalent is generally recommended.

Grade 3 pneumonitis is defined by radiographic infiltrates accompanied by moderate symptoms with hypoxia. Grade 4 pneumonitis is defined by life-threatening respiratory failure. Patients with grade 3–4 pneumonitis should be admitted to the hospital. Appropriate diagnostic examinations, including bronchoscopy, should be performed to exclude infections or other etiologies before starting more aggressive therapies. Corticosteroids should be equivalent to intravenous (methyl)prednisolone at a dose of 2–4 mg/kg/day. Administration of empiric antibiotics may be considered. If symptoms or respiratory conditions improve to baseline, corticosteroids should be tapered over at least 6 weeks. However, if respiratory conditions worsen after 48 h, additional immunosuppressive or immunomodulatory agents should be considered, including intravenous immunoglobulin (IVIG), infliximab, cyclophosphamide, tacrolimus, and mycophenolate mofetil (see following sections). In these settings (grades 3–4), ICIs should be discontinued.

### 6.1. Corticosteroids

The standard treatment for ICI-pneumonitis is systemic corticosteroids, as described above, based on drug-related interstitial pneumonitis treatment [63]. Corticosteroids have inhibitory effects on a broad range of immune responses via genomic and rapid non-genomic pathways [64]. Genomic effects include blocking promotor sites of proinflammatory genes, such as interleukin (IL)-1, inducing anti-inflammatory genes such as I-kappa-B-alpha, IL-10, alpha-2-macroglobulin, and inhibition of the synthesis of almost all inflammatory cytokines by competing with the function of nuclear factor-kappa-B and activator protein-1. Non-genomic effects include the inhibition of inflammatory cytokines via post-translational regulation, and increased degradation of mRNAs encoding IL-1, IL-2, IL-6, IL-8, and tumor necrosis factor. Although large clinical trials have not yet been conducted to investigate the efficacy of corticosteroids on ICI-pneumonitis, it is reasonable to use corticosteroids, considering the possible pathomechanisms of ICI-pneumonitis. Two major problems remain: A broad range of adverse effects and corticosteroid-refractory ICI-pneumonitis.

### 6.2. Infliximab

Infliximab is an anti-human TNF-α monoclonal antibody approved for rheumatoid arthritis, ulcerative colitis, and ankylosing spondylitis. Infliximab has been reported in several clinical trials by virtue of its usefulness in steroid-refractory, ICI-related colitis [65,66]. Regarding ICI-pneumonitis, there have been cases where infliximab was effective [40,67]. However, the current recommendation for the use of infliximab is largely from colitis data and levels of TNF-α were not elevated in BALF from ICI-pneumonitis compared with healthy controls [68]. Further evidence is warranted.

### 6.3. Intravenous Immunoglobulin (IVIG)

IVIG has been used for decades in various infections and autoimmune diseases. IVIg exerts an anti-inflammatory effect by suppressing the actions of autoantibodies and cytokines, and has been used to treat irAEs, such as ICI-related myasthenia gravis or dermatomyositis [51,69]. A case report describes severe, steroid-refractory ICI-pneumonitis successfully treated with IVIG [70]. Although no large datasets of IVIG for ICI-pneumonitis exist, IVIG is an attractive option, especially for patients with ICI-pneumonitis in cases where a comorbid infection is suspected, given that IVIG does not suppress innate and humoral responses to infection, unlike other immunosuppressive agents. Thus, IVIG treatment is proposed as a secondary option for severe cases of ICI-pneumonitis [52,61].

### 6.4. Tocilizumab

Tocilizumab is an anti-IL-6 receptor antibody with adequate efficacy and safety for rheumatoid arthritis [71]. Levels of IL-6 were elevated in BALF from ICI-pneumonitis compared with healthy controls (126.0 vs. 1.9 pg/mL *p* = 0.044) [68]. Recently, tocilizumab efficacy has been reported from a single center for patients with irAEs that were refractory to initial steroid therapy [72]. Of 87 patients who developed irAEs, 34 were refractory to steroids and required tocilizumab in addition to steroids. Among 34 patients, 12 developed grade 3/4 pneumonitis. Twenty-seven patients (approximately 80%) showed clinical improvement. Thus, tocilizumab has potential as a future treatment option for steroid-refractory ICI-pneumonitis.

### 6.5. Cyclophosphamide and Tacrolimus

Cyclophosphamide is an alkylating agent used to treat patients with various types of cancer. Tacrolimus is a calcineurin inhibitor, targeting T cell activation, and it has been used in immunosuppressive regimens for solid organ transplantation. Both cyclophosphamide and tacrolimus can also be used for myositis-associated ILDs [73]. The potential effectiveness of triple therapy of high-dose corticosteroids, calcineurin inhibitors, and cyclophosphamide has been reported for clinically amyopathic dermatomyositis-associated ILD, which is one of the most severe acute ILDs [74]. Similarly, a case report of successful treatment with triple therapy (high-dose steroids, tacrolimus, and cyclophosphamide) for severe, steroid-refractory ICI-pneumonitis has been published [75].

### 6.6. Vasoactive Intestinal Peptide (VIP)

Vasoactive intestinal peptide (VIP) is a hormone found in the pancreas, intestine, central nervous system, and lymphoid tissues, with many actions in the body, such as anti-inflammation activities [76]. In older literature, VIP inhalation increased alveolar regulatory T cells, decreased inflammatory cytokines, and improved clinical symptoms in sarcoidosis [77]. Bjorn et al. reported that VIP inhalation improved ICI-pneumonitis caused by pembrolizumab in advanced melanoma [78]. Importantly, VIP inhalation was not associated with toxic effects [78], in contrast to corticosteroid or other immunosuppressive treatments. Further, VIP inhalation did not influence lymphocyte subtypes in peripheral blood, suggesting the maintenance of systemic anti-tumor effects. Therefore, VIP inhalation has the potential to become a standard treatment for ICI-pneumonitis. Further study of VIP efficiency against ICI-pneumonitis is warranted.

### 6.7. Rechallenge of ICIs

Clinically, physicians often wonder whether ICIs could be re-administered to patients who had developed ICI-pneumonitis. As mentioned above, in settings with grades 3 to 4, guidelines recommend discontinuing ICIs permanently [52,60]. On the other hand, in settings with grade 2, ICIs can be resumed once ICI-pneumonitis returns to grade 1 or less [52,60]. Dolladille et al. reported on 6123 cases of the ICI rechallenge after an occurrence of adverse events [79]. Among the 6123 cases, 452 were involved in an informative rechallenge. One hundred and thirty (28.8%) of them had the same adverse events as the first time. ICI-pneumonitis was especially associated with a higher recurrence rate (OR 2.26, 95% CI: 1.18–4.32) compared with other irAEs [79]. Therefore, whether an ICI rechallenge could benefit patients depends on the case. Physicians should consider cancer status, comorbidities, and performance status of patients, and should inform patients of these occurrence rates in determining whether to rechallenge ICIs.

## 7. Prognosis

As introduced, ICI-pneumonitis represents the most common fatal irAE from anti-PD-1/PD-L1 monotherapy, accounting for 35% of anti-PD-1/PD-L1-related deaths [12]. In patients with non-small cell lung cancer, who had developed ICI-pneumonitis grade 3 or 4, accounted for 48.7% (19/39) of cases and grade 5 accounted for 12.8% (5/39) [67]. A multicenter retrospective survey revealed that grade 3 or 4 accounted for 45% (29/64) and grade 5 accounted for 9.4% (6/64) of cancer patients who developed ICI-pneumonitis [7]. In our study, grade 5 accounted for 7.6% (1/13) [24]. Thus, ICI-pneumonitis can be severe and fatal, so physicians should not underestimate ICI-pneumonitis.

## 8. Concluding Remarks

ICI-pneumonitis is a potentially severe irAE presenting varied symptoms and radiological manifestations. Real-world evidence shows a higher incidence of ICI-pneumonitis than previously reported in clinical trials. Diagnostic and management approaches to ICI-pneumonitis have been improved, with several society guidelines and publications. Several treatment options have been reported to treat steroid-refractory ICI-pneumonitis. Recent studies have uncovered certain pathobiological mechanisms of ICI-pneumonitis. Cellular immunity mediated by ICI-activated T cells is considered highly involved in ICI-pneumonitis. The imbalance between effector T cells and Tregs, the disequilibrium of PD-1/PD-L1 *cis*-regulation in autoimmune tolerance, and anti-CD74 autoantibodies may underlie the pathogenesis. VIP inhalation has the potential to become standard therapy for ICI-pneumonitis. However, further investigations are needed to investigate detailed biological mechanisms of ICI-pneumonitis and to identify optimal management strategies for ICI-pneumonitis, especially for severe cases.

## Figures and Tables

**Figure 1 biomedicines-09-01484-f001:**
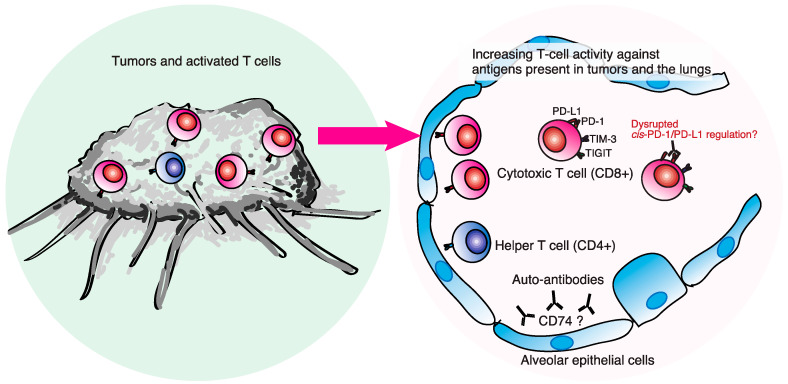
Possible pathobiological mechanisms underlying immune-checkpoint inhibitor-related pneumonitis. Cellular immunity mediated by T cells is thought to be highly involved. T-cell-dominated lymphocytosis is present in BALF from ICI-pneumonitis. CD8+ T cells in BALF express immune-checkpoint proteins (PD-1+TIM-3+), characteristic of tumor-infiltrating T lymphocytes. A substantial number of T cell clones in BALF were identical with T cells in the peritumoral lesion. These facts support the hypothesis that ICI-pneumonitis may be induced by ICI-activated T cells recognizing self-peptides or shared epitopes between tumors and the lungs. Moreover, the proportion of PD-1+PD-L1+ cells among CD8+ T cells in BALF was correlated with the severity of ICI-pneumonitis by anti-PD-1 therapies, suggesting that the disruption of cis-PD-1/PD-L1 regulation by anti-PD-1/PD-L1 therapies can induce aberrant activation of self-reacting T cells, resulting in severe ICI-pneumonitis. Autoantibodies such as anti-CD74 antibodies may also be involved in the pathogenesis of ICI-pneumonitis.

**Figure 2 biomedicines-09-01484-f002:**
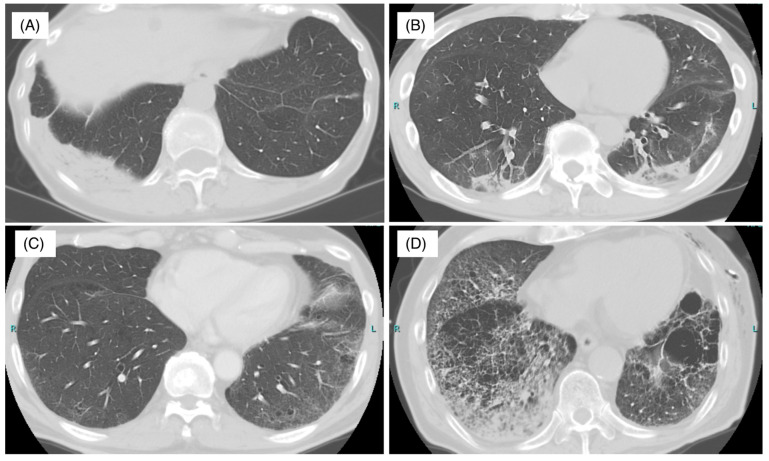
Radiologic features of immune-checkpoint inhibitor-related pneumonitis. (**A**) Organizing pneumonia (OP) pattern in a patient with stage IV renal cell carcinoma treated with nivolumab. An axial chest CT image obtained 4 months after nivolumab therapy shows an infiltration shadow on the peripheral pleural side, predominantly in the lower lung field. (**B**) A nonspecific interstitial pneumonia (NSIP) pattern in a patient with stage IV bladder cancer treated with nivolumab. An axial chest CT image obtained 11 months after nivolumab therapy shows bilateral lower lobe ground-glass opacities and reticular opacities with regions of subpleural sparing. (**C**) A hypersensitivity pneumonitis (HP) pattern in a patient with stage IV lung adenocarcinoma treated with pembrolizumab. An axial chest CT image obtained 5 months after pembrolizumab therapy shows diffuse, subtle ground-glass opacities. (**D**) A diffuse alveolar damage (DAD) pattern in a patient with stage IV lung adenocarcinoma treated with pembrolizumab. An axial chest CT image obtained 5 months after pembrolizumab therapy shows diffuse ground-glass opacities and consolidation. All images were collected at Kyushu University.

**Table 1 biomedicines-09-01484-t001:** Differential diagnosis of immune-checkpoint inhibitor (ICI)-pneumonitis.

Differential Diagnosis	Considerable Diseases	Examinations
Infections	Bacteria pneumonia, Pneumocystis jirovecii pneumonia, Flare-ups of tuberculosis	Microbiological tests Bronchial endoscopy
Heart failure	Myocardial infarction, Pulmonary edema, Myocarditis	Cardiac assessment
Cancer progression	Carcinomatous lymphangiosis, Pulmonary tumor embolism	Radiologic findings, Biopsy
Drug-induced pneumonitis	Induced by drugs other than ICIs	Check drug lists
Extra-pulmonary disease	Myasthenia gravis, Dermatomyositis/polymyositis	Muscle investigation

**Table 2 biomedicines-09-01484-t002:** Grading of immune-checkpoint inhibitor (ICI)-related pneumonitis and required treatment.

Grade	Grade 1	Grade 2	Grade 3	Grade 4	Grade 5
Symptoms	Asymptomatic	Symptomatic	Severe symptoms, Requiring oxygen therapy	Life-threatening respiratory failure	Death
Intervention	Careful observation Withhold ICIs	Withhold ICIs 0.5–1 mg/kg PSL	Discontinue ICIs 2–4 mg/kg PSL Empirical antibiotics	Discontinue ICIs 2–4 mg/kg PSL, Empirical antibiotics Immunosuppressive agents, IVIG	-

## Data Availability

The data in this study are available from corresponding author on reasonable request.

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
