# Peer review of "Insights into Potential Pathogenesis and Treatment Options for Immune-Checkpoint Inhibitor-Related Pneumonitis"

_biomedicines, 2021, doi:10.3390/biomedicines9101484_

Round 1

Reviewer 1 Report

The review titled “Immune-checkpoint inhibitor-related pneumonitis” by Ando et al. discusses a crucial topic and is well written. They have discussed that Immune checkpoint inhibitors (ICIs) targeting CTLA-4, PD-1, and PD-L1 have emerged as novel therapeutic possibilities for different cancers. IgE checkpoint inhibitory receptors or the ligands and impede inhibitory signals to boosting the immunological response to malignancies. However, Pneumonitis caused by immune checkpoint inhibitors (ICIs) is a potentially deadly immune-related side effect that may occur with these medications (irAE). The treatment may harm organs, including the lungs. This toxicity occurs in a small % of patients receiving anti-PD(L)1 monotherapy and a relatively higher % of patients getting anti-PD(L)1-based combos such as ipilimumab/nivolumab or non-small cell lung cancer (NSCLC). However, a tiny percentage of individuals may develop recurrent pneumonitis when re-challenged with ICIs. (ICI-pneumonitis) is a rare yet severe side effect that may be deadly. This review focuses on review current research on ICI-pneumonitis, focusing on pathogenesis and treatment.

Minor Comments

  1. The authors may include a few lines discussing the current status and future directions regarding; if there are different molecular phenotypes (Treg suppressive etc.) observed with ICI-pneumonitis that may be molecularly characterized?
  2. Risk stratification may help, and any future thoughts that can help researchers?
  3. The authors may include a few lines discussing; What can be the future management strategies? Moreover, how can researchers better study ICI-pneumonitis by using the right model system?
  4. Future immune targets may be discussed in the conclusion section.
  5. Figure 2: Any references for the same.

Author Response

1.    The authors may include a few lines discussing the current status and future directions regarding; if there are different molecular phenotypes (Treg suppressive etc.) observed with ICI-pneumonitis that may be molecularly characterized?

We appreciate the reviewer’s suggestion. Suresh et al. reported increased numbers of central memory T cells (high CD62L and low CD45RA expression), increased TNF-ahiIFNdhiCD8+ T cells, and a decreased Treg suppressive phenotype in BALF from ICI-pneumonitis [27]. Presumably, these central memory phenotype with TNF-ahiIFNdhiCD8+ T cells are the same population detected as CD8+PD-1+TIM-3+ in the study reported by Suzuki et al. For better understanding of pathogenic mechanisms, it may be necessary to investigate the imbalance between effector T cells and suppressor T cells (Tregs), to assess the degree of involvement of PD-1/PD-L1 cis-regulation in the autoimmune tolerance, to discover self-antigens expressed in the lungs, and to assess the biological role of anti-CD74 autoantibodies. We have included these considerations in the main text.

2.    Risk stratification may help, and any future thoughts that can help researchers?

We agree with the reviewer’s comments. We have added the following text. 
“In particular, prior thoracic radiotherapy (odds ratio (OR) 3.33), prior lung disease (OR 2.82), and combination therapy (3.42) are identified as risk factors analyzed by multiple logistic regression in patients with malignancies treated with ICIs [42]. These facts suggest a hypothesis that lung damage by radiation, chemotherapy, existing lung diseases may result in denatured self-antigens in the lung, which will be attracting circulating T cells that are activated by ICIs. Such activated T cells may misidentify these denatured self-antigens as non-self, resulting in inducing ICI-pneumonitis.” 

3.    The authors may include a few lines discussing; What can be the future management strategies? Moreover, how can researchers better study ICI-pneumonitis by using the right model system?

In my opinion, vasoactive intestinal peptide (VIP) inhalation has potential to become a standard treatment for ICI-pneumonitis, since VIP inhalation was not associated with toxic effects (N. Engl. J. Med. 2020, 382, 2573–2574.) VIP inhalation differs significantly from corticosteroid or other immunosuppressive treatments. Furthermore, VIP inhalation did not influence lymphocyte subtypes in peripheral blood, suggesting maintenance of systemic anti-tumor effects.  Recently, machine learning has been employed to predict TCR binding specificities of antigens presented by class I major histocompatibility complexes using only the TCR sequence, antigen sequence, and class I major histocompatibility complex allele (Nature Machine Intelligence 2021 Epub ahead of print. DOI: 10.1038/s42256-021-00383-2). This method can be used to predict self-antigens in ICI-pneumonitis, which will unveil the real targets of ICI-pneumonitis. I have added this explanation to the main text.

4.    Future immune targets may be discussed in the conclusion section.

I have added text regarding future immune targets in the Conclusion section.

5.    Figure 2: Any references for the same.

All images were collected at Kyushu University. I have added this information.

Reviewer 2 Report

In this review, the authors summarized recent findings regarding immune-checkpoint inhibitor (ICI)-related-pneumonitis, with a focus on potential pathogenesis and treatment. The manuscript is well written, easy to understand and well organized. There are some concerns about their claim.

Major Comment

A major problem of this review is that it lacks a description of the re-administration of ICI for patients who present relatively mild drug-induced lung injury. In particular, I would like the authors to mention 1) the conditions under which re-administration of ICI is acceptable, and 2) the frequency of drug-induced lung injury induced by ICI after re-administration. Dolladille et al. (JAMA Oncol 6(6):865-871, 2020) reported on 6,123 cases of the ICI rechallenge after an occurrence of adverse reactions. In their report, adverse events occurred in 7.4% of patients after the rechallenge, and 28.8% of them had the same adverse events as the first time. Since we often wonder whether ICI should be re-administered to patients who have once developed mild pneumonitis, it would be helpful for the readers to have your views on these questions.

Minor comments

  1. Drug-induced lung injury showing a pattern of hypersensitivity pneumonitis is relatively unlikely to present with centrilobular nodules as in hypersensitivity pneumonitis caused by inhaled antigens. The description "diffuse centrilobular ground-glass nodules" in Figure 2C is difficult to understand from the image. Please change the legend of the corresponding figure or modify the image appropriately.
  2. Page 4, 1st paragraph: This section discusses the differential diagnosis of ICI-associated pneumonia, which is of great clinical value. If possible, please create a Table to highlight the differential diagnosis.

Author Response

Major Comment

A major problem of this review is that it lacks a description of the re-administration of ICI for patients who present relatively mild drug-induced lung injury. In particular, I would like the authors to mention 1) the conditions under which re-administration of ICI is acceptable, and 2) the frequency of drug-induced lung injury induced by ICI after re-administration. Dolladille et al. (JAMA Oncol 6(6):865-871, 2020) reported on 6,123 cases of the ICI rechallenge after an occurrence of adverse reactions. In their report, adverse events occurred in 7.4% of patients after the rechallenge, and 28.8% of them had the same adverse events as the first time. Since we often wonder whether ICI should be re-administered to patients who have once developed mild pneumonitis, it would be helpful for the readers to have your views on these questions.

We truly appreciate the reviewer’s suggestion and we completely agree. We have created an additional sub-section regarding rechallenge of ICIs as follows:

“Clinically, physicians often wonder whether ICIs could be re-administered to patients who had developed ICI-pneumonitis. As mentioned above, in settings with grade 3 to 4, guidelines recommend discontinuing ICIs permanently [59][61]. On the other hand, in settings with grade 2, ICIs can be resumed once ICI-pneumonitis returns to grade 1 or less [59][61]. Dolladille et al. reported on 6,123 cases of the ICI rechallenge after an occurrence of adverse events [81]. Among the 6,123 cases, 452 were informative rechallenge. One hundred and thirty (28.8%) of them had the same adverse events as the first time. ICI-pneumonitis was especially associated with a higher recurrence rate (OR 2.26, 95% CI: 1.18–4.32) compared with other irAEs [81]. Therefore, whether ICI rechallenge could benefit patients depends on the case. Physicians should consider cancer status, comorbidities, and performance status of patients, and should inform patients of these occurrence rates in determining whether to rechallenge ICIs.”

Minor comments

Drug-induced lung injury showing a pattern of hypersensitivity pneumonitis is relatively unlikely to present with centrilobular nodules as in hypersensitivity pneumonitis caused by inhaled antigens. The description "diffuse centrilobular ground-glass nodules" in Figure 2C is difficult to understand from the image. Please change the legend of the corresponding figure or modify the image appropriately.

We have changed the legend to read, “diffuse, subtle ground-glass opacities”.

Page 4, 1st paragraph: This section discusses the differential diagnosis of ICI-associated pneumonia, which is of great clinical value. If possible, please create a Table to highlight the differential diagnosis.

We have added a Table to highlight the differential diagnosis.

Reviewer 3 Report

Authors described review article on immune-checkpoint inhibitor (ICI)-related pneumonitis. They summarized findings of pathogenesis, risk factors, clinical presentations, and therapy for ICI-related pneumonitis in their manuscript.

(Major comments)

  1. Several review articles of ICI-related pneumonitis are already found in Medline. The content of this manuscript is very similar to previous review papers. What do they would like to focus on their report? This should be described in the introduction. The title could be different from those of previous review articles and changed along with what they would like to focus on.
  2. They suggest that ICI-related pneumonitis is serious and can be fatal; however, there is no description on the prognosis of this disease. They can write down these findings by adding new number (section #6. etc.) before “conclusion remarks”.
  3. The guideline for ICI-related pneumonitis therapy is described in section #5. The main therapeutic strategy for ICI-related pneumonitis therapy is corticosteroid therapy. They should add the section of corticosteroid therapy (as new # 5.1) and describe in detail including mechanism and evidence of therapeutic effect of corticosteroid.

(Minor comments)

  1. The usefulness of infliximab and tocilizumab is described. Are there any reports of elevated levels of TNF-α and IL-6 in the bronchoalveolar lavage fluid or lung specimen from patients with ICI-related pneumonitis? They can add these findings in each section (old # 5.1. and # 5.2.).

Author Response

Several review articles of ICI-related pneumonitis are already found in Medline. The content of this manuscript is very similar to previous review papers. What do they would like to focus on their report? This should be described in the introduction. The title could be different from those of previous review articles and changed along with what they would like to focus on.

We agree with the reviewer. We have changed the title and the introduction to focus on potential pathogenesis and treatment options.

They suggest that ICI-related pneumonitis is serious and can be fatal; however, there is no description on the prognosis of this disease. They can write down these findings by adding new number (section #6. etc.) before “conclusion remarks”.

We agree with the reviewer. We have added a subsection regarding prognosis as follows.

7. Prognosis

As introduced, ICI-pneumonitis represents the most common fatal irAE from anti-PD-1/PD-L1 monotherapy, accounting to 35% of anti-PD-1/PD-L1-related deaths [12]. In patients with non-small cell lung cancer, who had developed ICI-pneumonitis grade 3 or 4, accounted for 48.7% (19/39) of cases and grade 5 accounted for 12.8% (5/39) [69]. A multicenter retrospective survey revealed that grade 3 or 4 accounted for 45% (29/64) and grade 5 accounted for 9.4% (6/64) of cancer patients who developed ICI-pneumonitis [82]. In our study, grade 5 accounted for 7.6% (1/13) [24]. Thus, ICI-pneumonitis can be severe and fatal, so physicians should not underestimate ICI-pneumonitis.”

The guideline for ICI-related pneumonitis therapy is described in section #5. The main therapeutic strategy for ICI-related pneumonitis therapy is corticosteroid therapy. They should add the section of corticosteroid therapy (as new # 5.1) and describe in detail including mechanism and evidence of therapeutic effect of corticosteroid.

We have added a subsection regarding corticosteroids (new section #6.1).

6.1. Corticosteroids

The standard treatment for ICI-pneumonitis is systemic corticosteroids, as described above, based on drug-related interstitial pneumonitis treatment [64]. Corticosteroids have inhibitory effects on a broad range of immune responses via genomic and rapid non-genomic pathways [65]. Genomic effects include blocking promotor sites of proinflammatory genes, such as interleukin (IL)-1, inducing anti-inflammatory genes such as I-kappa-B-alpha, IL-10, alpha-2-macroglobulin, inhibition of synthesis of almost all inflammatory cytokines by competing with the function of nuclear factor-kappa-B and activator protein-1. Non-genomic effects include inhibition of inflammatory cytokines via post-translational regulation, increased degradation of mRNAs encoding IL-1, IL-2, IL-6, IL-8, and tumor necrosis factor. Although large clinical trials have not been conducted yet to investigate the efficacy of corticosteroids on ICI-pneumonitis, it is reasonable to use corticosteroids, considering the possible pathomechanisms of ICI-pneumonitis. Two major problems remain: a broad range of adverse effects and corticosteroid-refractory ICI-pneumonitis.

(Minor comments)

The usefulness of infliximab and tocilizumab is described. Are there any reports of elevated levels of TNF-α and IL-6 in the bronchoalveolar lavage fluid or lung specimen from patients with ICI-related pneumonitis? They can add these findings in each section (old # 5.1. and # 5.2.).

Levels of IL-6 were elevated in BALF from ICI-pneumonitis compared with healthy controls (126.0 vs 1.9 pg/mL p=0.044) [73]. In contrast, levels of TNF-α were not elevated in BALF from ICI-pneumonitis compared with healthy controls in the same study [73]. I could not find any sources that show TNF- α elevation in BALF or lung specimens from ICI-pneumonitis. I have added this information in sections 6.1 and 6.2.

Round 2

Reviewer 3 Report

None.